# Coarse-to-fine Animal Pose and Shape Estimation

**Chen Li**     **Gim Hee Lee**
Department of Computer Science
National University of Singapore
`{lic, gimhee.lee}@comp.nus.edu.sg`

## Abstract

Most existing animal pose and shape estimation approaches reconstruct animal meshes with a parametric SMAL model. This is because the low-dimensional pose and shape parameters of the SMAL model makes it easier for deep networks to learn the high-dimensional animal meshes. However, the SMAL model is learned from scans of toy animals with limited pose and shape variations, and thus may not be able to represent highly varying real animals well. This may result in poor fittings of the estimated meshes to the 2D evidences, *e.g.* 2D keypoints or silhouettes. To mitigate this problem, we propose a coarse-to-fine approach to reconstruct 3D animal mesh from a single image. The coarse estimation stage first estimates the pose, shape and translation parameters of the SMAL model. The estimated meshes are then used as a starting point by a graph convolutional network (GCN) to predict a per-vertex deformation in the refinement stage. This combination of SMAL-based and vertex-based representations benefits from both parametric and non-parametric representations. We design our mesh refinement GCN (MRGCN) as an encoder-decoder structure with hierarchical feature representations to overcome the limited receptive field of traditional GCNs. Moreover, we observe that the global image feature used by existing animal mesh reconstruction works is unable to capture detailed shape information for mesh refinement. We thus introduce a local feature extractor to retrieve a vertex-level feature and use it together with the global feature as the input of the MRGCN. We test our approach on the StanfordExtra dataset and achieve state-of-the-art results. Furthermore, we test the generalization capacity of our approach on the Animal Pose and BADJA datasets. Our code is available at the project website[1].

## 1   Introduction

Animals play an important role in our everyday life and the study of animals has many potential applications in zoology, farming and ecology. Although great success has been achieved for modeling humans, progress for the animals counterpart is relatively slow mainly due to the lack of labeled data. It is almost impossible to collect large scale 3D pose data for animals, which prevents the direct application of many existing human pose estimation techniques to animals. Some works solve this problem by training their model with synthetic data [4, 13, 33]. However, generating realistic images is challenging, and more importantly, the limited pose and shape variations of synthetic data may result in performance drop when applied to real images. Other works avoid the requirement of 3D ground truth by making use of 2D weak supervision. For example, [2, 12] render the estimated mesh or project the 3D keypoints onto the 2D image space, and then compute the losses based on ground truth silhouettes or 2D keypoints. This setting is more pragmatic since 2D animal annotations are more accessible. In view of this, we focus on learning 3D animal pose and shape with only 2D weak supervision in this work.

---

[1] `https://github.com/chaneyddtt/Coarse-to-fine-3D-Animal`

35th Conference on Neural Information Processing Systems (NeurIPS 2021).

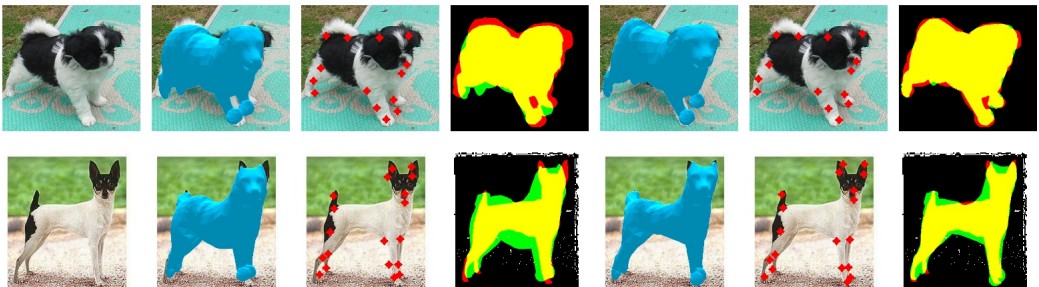

Figure 1: Some results of our coarse-to-fine approach. The first column is the input RGB images. The second to fourth columns are the estimated meshes in the camera view, the projected 2D keypoints and silhouettes from the coarse estimations. The fifth column to seventh columns are the estimated meshes, the projected 2D keypoints and silhouettes from the refined estimations. Note that The red and green regions in the fourth and seventh columns represent the ground truth and rendered silhouettes, respectively, and the yellow region is the overlap between them.

Reconstruction of the 3D mesh from a single RGB image with only 2D supervision is challenging because of the articulated structure of animals. Most existing works [4, 2] rely on the parametric statistical body shape model SMAL [35] to reconstruct animal poses and shapes. Specifically, the low dimensional parameterization of the SMAL model makes it easier for deep learning to learn the high dimensional 3D mesh with 3,889 vertices for high-quality reconstruction. However, the shape space of the SMAL model is learned from 41 scans of toy animals and thus lacks pose and shape variations. This limits the representation capacity of the model and results in poor fittings of the estimated shapes to the 2D observations when applied to in-the-wild animal images, as illustrated in Figure 1. Both 2D keypoints (the left front leg in the first example and right ear in the second example) and silhouettes rendered from the coarse estimations do not match well with the ground truth annotations (rendered and ground truth silhouettes are denoted with green and red regions, and the yellow region is the overlap between them). To mitigate this problem, we design a two-stage approach to conduct a coarse-to-fine reconstruction. In the coarse estimation stage, we regress the shape and pose parameters of the SMAL model from an RGB image and recover a coarse mesh from these parameters. The coarse mesh is then used as a starting point by a MRGCN to predict a per-vertex deformation in the refinement stage. This combination of SMAL-based and vertex-based representations enable our network to benefit from both parametric and non-parametric representations. Some refined examples are shown in Figure 1, where both keypoint locations and overall shapes are improved.

To facilitate the non-parametric representation in the refinement stage, we consider a mesh as a graph structure and use a GCN [15] to encode the mesh topology (i.e., the edge and face information). Existing animal mesh reconstruction estimation works [2, 12, 33] rely on a global image feature to recover 3D meshes. However, we observe that the global feature fails to capture local geometry information that is critical to recover shape details. To solve this problem, we introduce a local feature extractor to retrieve a per-vertex feature and use it together with the global feature as the input of each node in the MRGCN. The combination of the vertex-level local feature and the image-level global feature helps to capture both overall structure and detailed shape information. Moreover, the representation power of traditional GCNs is limited by the small receptive field since each node is updated by aggregating features of neighboring nodes. To mitigate this, we design our MRGCN as an encoder-decode structure. Specifically, the downsampling and upsampling operations in the encoder-decoder structure enables a hierarchical representation of the mesh data, where features of different mesh resolutions can be exploited for mesh refinement.

Our main contributions can be summarized as follows: 1) We propose a coarse-to-fine approach that combines parametric and non-parametric representations for animal pose and shape estimation. 2) We design an encoder-decoder structured GCN for mesh refinement, where both image-level global feature and vertex-level local feature are combined to capture overall structure and detailed shape information. 3) We achieve state-of-the-art results for animal pose and shape estimation on three animal datasets.

## 2   Related work

We review works on pose and shape estimation for both human and animals since they share similar ideas. We mainly discuss approaches that reconstruct 3D meshes which is the focus of this work.

**Human pose and shape estimation**   We first review model-based human pose estimation. This line of works are based on a parametric model of human body, such as SMPL [20] or SCAPE [1], to reconstruct 3D human meshes. Bogo *et al.*[5] first propose a fully automatic approach SMPLify, which fits the SMPL model to 2D locations estimated by an off-the-shelf 2D keypoint detector. The shapes estimated by SMPLify are highly underconstrained although strong priors are added during the fitting process. To improve this, subsequent works incorporate more information during optimization, such as silhouettes [19] or multi-view videos [10]. The drawback of the optimization-based approaches is that the iterative optimization process is quite slow and tends to fail when the 2D keypoints or silhouettes are noisy. On the other hand, regression-based approaches directly regress the body model parameters from an input image, which is more computationally efficient than optimization-based approaches. Kanazawa *et al.*[11] introduce an end-to-end framework for recovering 3D mesh of a human body by using unpaired 2D keypoints and 3D scans. Pavlakos *et al.*[23] propose to employ a differentiable renderer to project the generated 3D meshes back to 2D image space such that the network can be trained with 2D supervision. Other intermediate representations such as semantic segmentation map [22] or IUV map [31] are also introduced in order to improve the robustness. Recently, Kolotouros *et al.*[16] propose SPIN, where the optimization process is incorporated into the network training such that strong model-based supervision can be exploited.

It has been observed that directly regressing model parameters is quite challenging because of the representation issues of 3D rotation [28, 17, 7, 32, 8]. Hence, some works focus on model-free based human pose and shape estimation, where they do not directly regress the model parameters. Varol *et al.*[28] proposed BodyNet, which estimates volumetric representation of 3D human with a Voxel-CNN. Kolotouros *et al.*[17] design a graph convolutional human mesh regression model to estimate vertex locations from a global image feature. Choi *et al.*[7] propose a graph convolutional system that recovers 3D human mesh from 2D human pose. We also apply GCN in our approach to refine the initial coarse estimation. However, we do not rely on 3D supervision as done in [17, 7], and our mesh refinement GCN has a different architecture. Our work is also related to PIFu [25] in terms of making use of local features to better capture shape details. Instead of using a complete non-parametric representation as in [25], we combine the vertex-based representation with the SMAL-based representation such that our network can be weakly supervised with 2D annotations.

**Animal pose and shape estimation**   Animal pose estimation is less explored compared to human pose estimation mainly because of the lack of large scale datasets. It is almost impossible to collect thousands of 3D scan of animals as has been done in SMPL [20]. The first parametric animal model SMAL is recently proposed by Zuffi *et al.*[35], which is learned from 41 scans of animal toys. The SMAL model is further improved in [34] by fitting the model to multiple images such that it can generalize to unseen animals. In the view of the lack of animal data, synthetic datasets are generated in [33, 13, 4] to train their networks. However, there is domain gap between the synthetic and real images in terms of appearance, pose and shape variations. Although the domain shift problem can be mitigated by using a depth map [13] or silhouette[4] as the input, these information are not always available in practice. More recently, Biggs *et al.*[2] collect 2D keypoint and silhouette annotations for a large scale dog dataset and train their network with those 2D supervision. The authors observe that the SMAL model failed to capture the large shape variations of different dog breeds, and propose to incorporate an EM algorithm into training to update the shape priors. However, it is not guaranteed that the priors are valid shapes since they are computed from the network predictions. We also work on the dog category following [2] in this work, and we overcome the limited representation capacity of the SMAL model by combining it with a vertex-based representation.

## 3   Our method

We propose a two-stage approach for coarse-to-fine animal pose and shape estimation. The overall architecture is illustrated in Figure 2. The coarse estimation stage is based on the SMAL model, where we directly regress the pose, shape and translation parameters from the input image. We

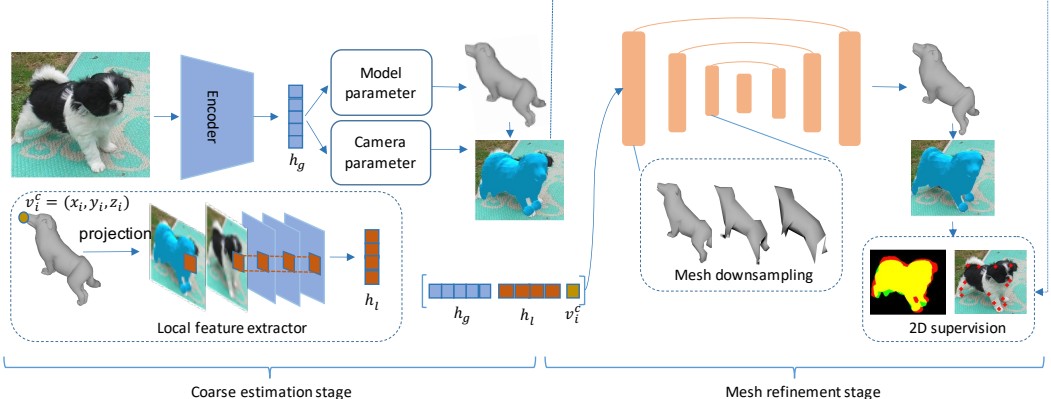

Figure 2: Overall architecture of our network. Our network consists a coarse estimation stage and a mesh refinement stage. The SMAL model parameters and camera parameter are regressed from the input image in the first stage for coarse reconstruction. This coarse mesh is further refined by an encoder-decoder structured GCN in the second stage, where the initial node feature consists of a global feature, a local feature and the initial 3D coordinate.

concurrently estimate the camera parameter to project the estimated 3D meshes onto the image space for the network supervision with the 2D annotations. Due to the limited representation capacity of the SMAL model, the coarse estimations may not fit well to the 2D evidences. We mitigate this problem by further refinement with a MRGCN in the second stage. The input feature for each node in the MRGCN consists of both image-level global feature and vertex-level local feature retrieved by a local feature extractor. The MRGCN is designed as an encoder-decoder structure, where the mesh downsampling and upsampling operations enable a hierarchical representation of the mesh data. Similar to the coarse estimation stage, the mesh refinement stage is also supervised with the 2D annotations.

## 3.1 Model-based coarse estimation

We use a parametric representation based on the SMAL model in the first stage. The SMAL model $\mathcal{M}(\beta, \theta, \gamma)$ is a function of shape $\beta$, pose $\theta$ and translation $\gamma$. The shape parameter $\beta$ is a vector of the coefficients of a PCA shape space learned from scans of animal toys. The pose parameter $\theta \in \mathbb{R}^{N \times 3}$ is the relative rotation of joints expressed with Rodrigues vectors, where we use $N = 35$ joints in the kinematic tree following [33, 2]. The translation $\gamma$ is a global translation applied to the root joint. Given a set of model parameters $\Theta = \{\beta, \theta, \gamma\}$, the SMAL model returns 3D mesh vertices $V = M(\beta, \theta, \gamma)$, where $V \in \mathbb{R}^{C \times 3}$ and $C = 3,889$ is the number of animal mesh vertices. The body joints of the model is defined as a linear combination of the mesh vertices $J \in \mathbb{R}^{N \times 3} = \mathcal{W} \times V$, where $\mathcal{W}$ is a linear regressor provided with the SMAL model.

As illustrated in the coarse estimation stage of Figure 2, we first extract a global feature $\mathbf{h}_g$ from the input image with an encoder. The global feature is fed into separate multi-layer perceptron (MLP) heads to predict the SMAL model parameters $\Theta' = \{\beta', \theta', \gamma'\}$ and the camera parameter $f$. Following [33, 2], we use independent branches for the translation in the camera frame $\gamma'_{xy}$ and depth $\gamma'_z$. The 3D meshes and joints can then be computed from the estimation parameters as:

$$V_c = \mathcal{M}(\beta', \theta', \gamma'), \qquad J_{3D} = \mathcal{W} \times V_c. \tag{1}$$

We further project the predicted 3D joints onto 2D image space with the estimated camera parameter and supervise them with the ground truth 2D keypoints:

$$\mathcal{L}_{kp1} = \|J_{2D} - \Pi(J_{3D}, f)\|^2. \tag{2}$$

$J_{2D}$ and $\Pi(.,.)$ denote the ground truth 2D keypoints and the camera projection function, respectively.

Existing works apply the $L_1$ [33] or $L_2$ distance [2, 23] to compute the loss for silhouettes. As illustrated in Figure 3, we observe that the model trained with $L_1$ or $L_2$ loss tends to generate a 3D mesh that has a foreground region (green region) larger than that of the ground truth silhouette (red region) after rendering. We analyze that this may be due to the imbalance between foreground and

background pixels in the training images. Note that we crop each image according to a bounding box during preprocessing. This causes the foreground object to occupy a larger area of an image and thus an imbalance of more foreground than background pixels. The imbalance problem may cause the network to estimate a mesh with larger size or larger focal length such that there will be more foreground pixels in the rendered silhouette. To solve this problem, we adopt the Tversky loss [26] from the semantic segmentation task:

$$\mathcal{T}(P, G; \alpha, \beta) = \frac{|PG|}{(|PG| + \alpha|P \backslash G| + \beta|G \backslash P|)}, \tag{3}$$

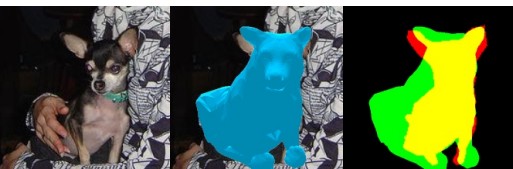

where $|PG|$ denotes foreground pixels present in both the predicted binary mask $P$ and the ground truth mask $G$. $|P \backslash G|$ denotes background pixels that are predicted as foreground and $|G \backslash P|$ vice versa, *i.e.* false positive and false negative predictions. $\alpha$ and $\beta$ are hyperparamters to control the weights for false positive and false negative predictions. By setting $\alpha$ to a value larger than $\beta$, the Tversky loss penalizes more on false positive predictions, which

Figure 3: Illustration of the rendered meshes occupying more pixels than 2D observation when applying $L_1$ loss for silhouette.

corresponds to the foreground pixels that are not present in the ground truth silhouettes. Our silhouette loss can thus be represented as:

$$\mathcal{L}_{\text{silh1}} = 1 - \mathcal{T}(S, \mathcal{R}(V_c, f), \alpha, \beta), \tag{4}$$

where $S$ represents the ground truth silhouette and $\mathcal{R}$ represents the mesh renderer, The network may predict unrealistic shapes and poses when trained with 2D supervision alone. In view of this, we encourage the predicted model parameters to be close to a prior distribution by computing the Mahalanobis distance between them:

$$\mathcal{L}_\beta = (\beta' - \mu_\beta)^\top \Sigma_\beta^{-1} (\beta' - \mu_\beta), \qquad \mathcal{L}_\theta = (\theta' - \mu_\theta)^\top \Sigma_\theta^{-1} (\theta' - \mu_\theta). \tag{5}$$

The prior distributions of the shape and pose are defined by the corresponding means $\{\mu_\beta, \mu_\theta\}$ and variances $\{\Sigma_\beta, \Sigma_\theta\}$ estimated from the SMAL training data. We also apply the pose limit prior [35], denoted as $\mathcal{L}_{\text{lim}}$, to enforce valid pose estimations. Finally, the overall loss function for the coarse estimation stage is computed as:

$$\mathcal{L}_{\text{st1}} = \lambda_{\text{kp1}} \mathcal{L}_{\text{kp1}} + \lambda_{\text{silh1}} \mathcal{L}_{\text{silh1}} + \lambda_\beta \mathcal{L}_\beta + \lambda_\theta \mathcal{L}_\theta + \lambda_{\text{lim}} \mathcal{L}_{\text{lim}}, \tag{6}$$

where $\lambda_{\text{kp1}}$, $\lambda_{\text{silh1}}$, $\lambda_\beta$, $\lambda_\theta$ and $\lambda_{\text{lim}}$ are the weights to balance different loss terms.

## 3.2  Model-free mesh refinement

As illustrated in the coarse estimations in Figure 1, the 3D meshes estimated from the first stage may not fit well to the 2D observations due to the limited representation capability of the SMAL model. We thus use the coarse estimation $V_c$ as a starting point and further refine it with our MRGCN. The GCN is able to encode the mesh topology and has been used in previous works [17, 7] for human pose and shape estimation. However, our MRGCN differs from [17, 7] in several aspects: 1) We incorporate a vertex-level local feature into the feature embedding of each node; 2) We design our MRGCN as an encoder-decoder structure with skip connections to exploit features of different scales; 3) We supervise our MRGCN with only 2D weak supervision.

**Combination of global and local features.**   The initial feature of each node in [17, 7] is either a combination of the global image feature and the initial 3D coordinate or a combination of the 3D coordinate and the corresponding 2D joint location. In contrast, we introduce a vertex-level local feature to capture more detailed shape information. This is inspired by previous works [30, 29] with the exceptions that we do not need ground truth 3D annotations or camera parameter. The local feature extraction process is illustrated in the left bottom part of Figure 2. We first project each mesh vertex $v_i^c = (x_i, y_i, z_i) \in V_c$ onto 2D image space with the estimate camera parameter $p_i = \Pi(v_i^c, f)$, and then retrieve the features at $p_i$ from all feature maps of the encoder. These features are concatenated and used as our local feature $\mathbf{h}_l$. Finally, the input feature for each vertex is a combination of the global feature, the local feature and the initial 3D coordinates $\mathbf{h}_i^0 = [\mathbf{h}_g, \mathbf{h}_l, x_i, y_i, z_i]$. This combination of both global and local features helps to capture the overall structure and detailed shape information.

**Encoder-decoder structured GCN.** We define the graph on animal mesh as $\mathcal{G} = (V, A)$, where $V$ represents the mesh vertices and $A \in \{0, 1\}^{C \times C}$ represents an adjacency matrix defining edges of animal mesh. We adopt the graph convolutional layer $H = \tilde{A}XW$ proposed in [15], where $\tilde{A}$ is symmetrically normalized from $A$, $X$ is the input feature and $W$ represents the weight matrix. In the original GCN, features of each node are updated by averaging over features of neighboring nodes. This results in a small receptive field and limits the representation capacity. To mitigate this problem, we design our MRGCN as an encoder-decoder architecture to effectively exploit features of different mesh resolutions. Specifically, the graph encoder consists of consecutive graph convolutional layers and downsampling layers, and the decoder is a symmetry of the encoder architecture except that the downsampling layers are replaced with upsampling layers. Following [24], the mesh downsampling is performed by using a transformation $V_k = Q_d V_l$, where $Q_d \in \{0, 1\}^{k \times l}$ represents the transformation matrix, and $l, k$ are the number of mesh vertices before and after downsampling. The downsampling transformation matrix is obtained by contracting vertex pairs iteratively based on quadric matrices. We show an illustration of the downsampling process in the mesh downsampling block of Figure 2. Similarly, the upsampling is performed by $V_l = Q_u V_k$. Furthermore, inspired by the stacked hourglass network [21], we add skip connections between the encoder and decoder layers to preserve the spatial information at each resolution.

**Weakly supervised mesh refinement.** Given the coarse estimation from the first stage, our MRGCN predicts a per-vertex deformation based on the input feature:

$$\Delta v_i = \mathcal{F}(\mathbf{h}_i^0), \quad \text{where} \quad \mathbf{h}_i^0 = [\mathbf{h}_g, \mathbf{h}_l, x_i, y_i, z_i]. \tag{7}$$

The final refined mesh is obtained by:

$$V_f = V_c + \Delta V, \quad \text{where} \quad \Delta V = [\Delta v_1, \Delta v_2, ...., \Delta v_C]. \tag{8}$$

Similar to the first stage, we project the mesh vertices and 3D keypoints onto 2D image space with the estimated camera parameter $f$, and compute the 2D keypoints loss $\mathcal{L}_{\text{kp2}}$ and silhouette loss $\mathcal{L}_{\text{silh2}}$ in the same way as Eq. (2) and Eq. (4). Note that we do not refine the camera parameter in the second stage as we postulate that the camera parameter relies more on the global features, *e.g.* the global scale. The shape and pose prior in the first stage can not be used as we are directly regressing the vertex deformation. Without any constraint, the network is susceptible to large deformations in its predictions in order to align perfectly with the 2D evidences. This may result in unrealistic output meshes. In view of this problem, we add a Laplacian regularizer to prevent large deformations such that only fine-grained details are added upon the coarse estimation. Specifically, we encourage the Laplacian coordinates of the refined mesh to be the same as the coarse mesh by enforcing:

$$\mathcal{L}_{\text{lap}} = \sum_i \|\delta v_i^f - \delta v_i^c\|^2, \quad \text{where} \quad \delta v_i = v_i - \frac{1}{d_i} \sum_{j \in N(i)} v_j. \tag{9}$$

$v^f$ and $v^c$ represent the vertex in the refined and coarse meshes, respectively. $\delta v_i$ is the Laplacian coordinate of $v_i$, which is the difference between the original coordinate and the center of its neighbors in the mesh. Finally, the training loss for the refinement stage can be represented as:

$$\mathcal{L}_{\text{st2}} = \lambda_{\text{kp2}} \mathcal{L}_{\text{kp2}} + \lambda_{\text{silh2}} \mathcal{L}_{\text{silh2}} + \lambda_{\text{lap}} \mathcal{L}_{\text{lap}}, \tag{10}$$

where $\lambda_{\text{kp2}}$, $\lambda_{\text{silh2}}$ and $\lambda_{\text{lap}}$ are the weights for corresponding losses.

## 4 Experiments

**Network details.** The structure of the encoder in the first stage is similar to [33, 2]. We extract the features of an input image of size $224 \times 224 \times 3$ with a Resnet50 [9] module, which is followed by a convolutional layer with group normalization and leaky ReLU. We further add two fully connected layers to obtain the global feature. This global feature is fed into different MLP branches to estimate the SMAL model parameters $\Theta$ and the camera parameter $f$. All intermediate feature maps of the encoder are resized and concatenated for the local feature extraction of the refinement stage. The encoder of our MRGCN consists of consecutive graph convolutional blocks and downsampling layers, where the mesh graph is downsampled by a factor of 4 at each time. As in [17], the graph convolutional block is similar to the Bottleneck residual block in [9], where the $1 \times 1$ convolutions are replaced by per-vertex fully connected layers and the Batch Normalization is replaced by Group Normalization. Similarly, the decoder consists of consecutive graph convolutional blocks and upsampling layers to reconstruct 3D mesh in the original resolution. The skip connections are added between the encoder and decoder layers with a concatenation operation.

Table 1: IOU and PCK@0.15 scores for the StanfordExtra dataset. 'GT' and 'Pred' represent whether ground truth or predicted keypoints or silhouettes are used during test, and '-' represents that the corresponding information is not required. 'Ours-coarse' and 'Ours' represent the results of the coarse and refined estimations, respectively. (Best results in bold)

| Method | Keypoints | Silhouette | IOU | PCK@0.15 | | | | |
| --- | --- | --- | --- | --- | --- | --- | --- | --- |
| | | | | Avg | Legs | Tail | Ears | Face |
| 3D-M [35] | Pred | Pred | 69.9 | 69.7 | 68.3 | 68.0 | 57.8 | 93.7 |
| 3D-M | GT | GT | 71.0 | 75.6 | 74.2 | **89.5** | 60.7 | **98.6** |
| 3D-M | GT | Pred | 70.7 | 75.5 | 74.1 | 88.1 | 60.2 | 98.7 |
| 3D-M | Pred | GT | 70.5 | 70.3 | 69.0 | 69.4 | 58.5 | 94.0 |
| CGAS [4] | CGAS | Pred | 63.5 | 28.6 | 30.7 | 34.5 | 25.9 | 24.1 |
| CGAS | CGAS | GT | 64.2 | 28.2 | 30.1 | 33.4 | 26.3 | 24.5 |
| WLDO [2] | - | - | 74.2 | 78.8 | 76.4 | 63.9 | 78.1 | 92.1 |
| Ours-coarse | - | - | 72.5 | 77.0 | 75.9 | 55.3 | 76.1 | 89.8 |
| Ours | - | - | **81.6** | **83.4** | **81.9** | 63.7 | **84.4** | 94.4 |

**Training details.** We train our network in three stages. We first train the coarse estimation part with all losses in $\mathcal{L}_{st1}$ except the silhouette loss $\mathcal{L}_{silh1}$ for 200 epochs. We did not include the pose limit prior $\mathcal{L}_{lim}$ at this stage as we find the network tends to estimate invalid poses only when we apply the silhouette loss. Subsequently, the mesh refinement part is trained with the keypoints loss $\mathcal{L}_{kp2}$ for 10 epochs with the coarse estimation part kept frozen. Finally, we train the whole network with both $\mathcal{L}_{st1}$ and $\mathcal{L}_{st2}$ for 200 epochs. We do not include the silhouette loss in the first two stages because it has been observed [4, 2] that the silhouette loss can lead the network to unsatisfactory local minima if applied too early. Our model is implemented with Pytorch, and we use the Adam optimizer with a learning rate of $10^{-4}$ in the first two stages and of $10^{-5}$ in the third stage. The whole training process takes around 30 hours on a RTX2080Ti GPU.

## 4.1 Datasets and evaluation metrics.

We train our model on the StanfordExtra dataset [2], and test on the StanfordExtra, the Animal Pose [6] and the benchmark animal dataset of joint annotations (BADJA) [4] datasets.

**StanfordExtra dataset.** The StanfordExtra dataset is a newly introduced large-scale dog dataset with 2D keypoint and silhouette annotations. The images are taken from an existing dataset for fine-grained image categorization [14], which consists of 20,580 images of dogs taken in the wild and covers 120 dog breeds. Images with too few keypoints within the field-of-view are removed. Each dog-breed is divided into training and testing split with a ratio of $4 : 1$.

**Animal Pose and BADJA datasets.** The Animal Pose dataset is recently introduced for 2D animal pose estimation, and the BADJA dataset is derived from the existing DAVIS video segmentation dataset. We use these two datasets to test the generalization capacity of our network.

**Evaluation metrics.** Following previous works [33, 2], we use two evaluation metrics: 1) Percentage of Correct Keypoints (PCK), and 2) intersection-over-union (IOU). PCK computes the percentage of joints that are within a normalized distance to the ground truth locations, and we use it to evaluate the quality of the estimated pose. We normalize the joint distance based on the square root of 2D silhouette area and use a threshold of 0.15, which we denote as PCK@0.15. Following [2], we also evaluate PCK@0.15 on different body parts, including legs, tail, ears and face. The IOU measures the overlap between the rendered silhouettes and the ground truth annotations, and we use it to evaluate the quality of the estimated 3D shapes.

## 4.2 Results on the StanfordExtra dataset

We show the results of our approach and compare with existing works in Table 1. We take the results of the existing works from [2]. 3D-M [35] is an optimization-based approach that fits the SMAL model to the 2D keypoints and silhouette by minimizing an energy term. The optimization is run for each image, which means 2D keypoints and silhouettes are needed during test. We list the results of 3D-M when ground truth (GT) or predicted (Pred) keypoints and silhouette are used. CGAS [4] first predicts 2D keypoints from silhouette with their proposed genetic algorithm, denoted as CGAS,

Table 2: IOU and PCK@0.15 scores for the Animal Pose dataset. (Best results in bold)

| Method | Keypoints | Silhouette | IOU | PCK@0.15 | | | | |
| --- | --- | --- | --- | --- | --- | --- | --- | --- |
| | | | | Avg | Legs | Tail | Ears | Face |
| 3D-M [35] | Pred | Pred | 64.9 | 59.2 | 55.7 | 56.9 | 61.3 | 86.7 |
| WLDO [2] | - | - | 67.5 | 67.6 | 60.4 | **62.7** | 86.0 | 86.7 |
| Ours-coarse | - | - | 67.5 | 62.0 | 57.1 | 45.1 | 75.8 | 78.9 |
| Ours | - | - | **75.7** | **67.8** | **62.2** | 45.1 | **86.6** | **87.8** |

and then run an optimization step to match the 2D evidences. Both WLDO [2] and our approach are regression-based, hence do not need 2D keypoints or silhouette during test. We show PCK@0.15 and IOU for both coarse (ours-coarse) and refined (ours) estimations. As shown in Table 1, our approach outperforms the optimization-based approaches [35, 4] in terms of both IOU and average PCK@0.15 even when ground truth 2D annotations are used. We also outperform the regression-based approach [2] for all metrics, with a relative improvement of 10.0% for IOU and 5.8% for average PCK@0.15. The fact that the refined estimation has higher scores than the coarse estimation demonstrates the effectiveness of our coarse-to-fine strategy. This is also evident from the qualitative results in Row 1-3 of Figure 4, where both keypoint locations (*e.g.* keypoints on the right rear legs in the first row and on the face in the second row) and silhouettes are improved (more yellow pixels and less red or green pixels) after the refinement step.

## 4.3 Results on the Animal Pose and BADJA datasets

To test the generalization capacity, we directly apply our model trained on StanfordExtra to the Animal Pose and BADJA datasets. The results for the Animal Pose dataset are shown in Table 2 . We take the performance of 3D-M and WLDO from [2]. Note that the results of 3D-M is obtained by using the predicted keypoints and silhouettes for fair comparison. We can see that our approach outperforms the state-of-the-art approach [2] for all metrics, especially for the IOU score. This demonstrates the importance of the refinement step in improving the coarse estimations since the architecture of our coarse estimation stage is similar to WLDO except that we do not apply the EM updates. We show the qualitative results on this dataset in Row 4-6 of Figure 4.

We also evaluate on the BADJA dataset and compare with the state-of-the-art WLDO in Table 3. The results of WLDO are obtained by running the publicly available checkpoint. Note that there is large domain gap between the BADJA and the StanfordExtra datasets in terms of animal pose and camera viewpoint. Most images in the BADJA dataset are taken

Table 3: IOU and PCK@0.15 for the BADJA dataset.

| Method | IOU | PCK@0.15 | | | | |
| --- | --- | --- | --- | --- | --- | --- |
| | | Avg | Legs | Tail | Ears | Face |
| WLDO [2] | 65.0 | 48.6 | 40.4 | **78.2** | 55.2 | 76.5 |
| Ours-coarse | 59.6 | 42.5 | 33.7 | 57.5 | 63.4 | **79.2** |
| Ours | **72.0** | **54.1** | **47.6** | 76.1 | **66.2** | 74.4 |

from a video where a dog is running, and sometimes the viewpoint is from the back. For the StanfordExtra dataset, most images are taken from a front view and the dogs are in a standing, siting or prone pose. We show some examples of the BADJA dataset in Row 7-9 of Figure 4 for illustration. We can see from Table 3 that our approach still obtains reasonable scores given the domain gap between these two datasets, improving over WLDO by 10.8% for IOU and 11.3% for average PCK@0.15. We also notice that the performance gap between the coarse and fine estimations is larger than that of the other two datasets. This may be attributed to the domain shift of the input image, which causes the coarse estimation to be inaccurate. Nonetheless, this inaccuracy can be reduced after refinement since the refinement step does not directly rely on the input image. We can observe from the qualitative results (Row 7-8) that keypoints on the legs aligned much better after refinement. This also demonstrates the advantage of our coarse-to-fine approach in the presence of domain gap. We show failure cases in the last two rows of Figure 4, where the camera is looking at the back of the animal or there is severe occlusion.

## 4.4 Ablation study

Table 4 shows the results of the ablation studies on the StandfordExtra dataset. We remove one component from our full pipeline (Full) at each time to evaluate the corresponding contribution.

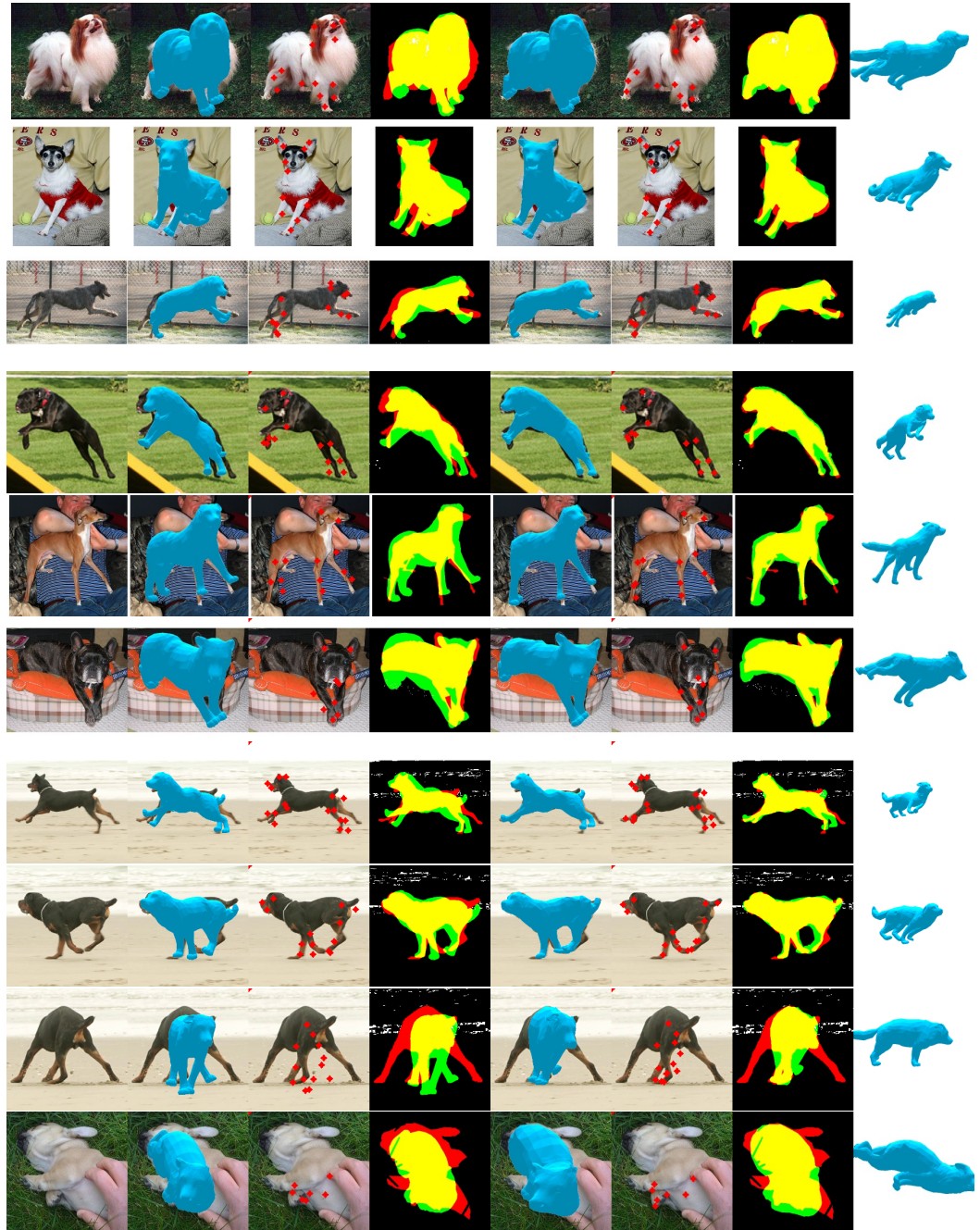

Figure 4: Qualitative results on the StanfordExtra, Animal Pose and BADJA datasets. The first column is the input images, the second to fourth columns and fifth to seventh columns are the coarse and refined estimations, respectively. The last column is the 3D mesh in another view.

We first evaluate our coarse-to-fine strategy by removing the MRGCN (-MR), which corresponds to the results of 'Ours-coarse' in the previous tables. We then test the contribution of the local features by removing it from the input of the MRGCN (-LF). This gives similar results with the coarse estimation, which means that the global feature alone is not able to improve the initial estimations. Our MRGCN is designed as

Table 4: Ablation study for each component.

| Method | IOU | PCK@0.15 | | | | |
|--------|-----|-----|------|------|------|------|
| | | Avg | Legs | Tail | Ears | Face |
| Full | **81.6** | **83.4** | **81.9** | 63.7 | **84.4** | **94.4** |
| -MR | 72.5 | 77.0 | 75.9 | 55.3 | 76.1 | 89.8 |
| -LF | 73.3 | 76.9 | 76.1 | 57.0 | 75.1 | 89.1 |
| -ED | 79.4 | 80.2 | 79.2 | 59.3 | 79.9 | 91.5 |
| -TL | 81.1 | 82.5 | 81.0 | **65.2** | 82.9 | 92.8 |

an encoder-decoder structure. We verify this design by replacing the structure with a traditional GCN (-ED), where the mesh vertices are processed in the same resolution across all layers. We also evaluate the contribution of the Tversky loss by replacing it with commonly used $L_1$ loss (-TL). We can see that the performance drops when each part is removed from the full pipeline. This verifies the contribution of each component.

## 5 Conclusion

We introduce a coarse-to-fine approach for 3D animal pose and shape estimation. Our two-stage approach is a combination of SMAL-based and vertex-based representations that benefits from both parametric and non-parametric representations. We design the mesh refinement GCN as an encoder-decoder structure, where the input feature of each node consists of both image-level and vertex-level features. This combination of global and local features helps to capture overall structure and detailed shape information. Our approach outperforms both optimization-based and regression-based approaches on the StanfordExtra dataset. Results on the Animal Pose and BADJA datasets verify the generalization capacity of our approach to unseen datasets.

## Limitations and potential negative societal impacts

There still exist challenging cases not handled by our work, such as scenarios where the camera looks at the back of the animal or with severe occlusions. A naive adoption of our approach to monitor animal health may result in potential negative consequences, *i.e.* the inaccurate shape reconstruction of the animals in this scenario can provide misleading information for the farmer or animal breeder. Nonetheless, the problem can be mitigated by collecting animal images with more variations of camera views and animal poses or by synthesizing novel images with existing datasets [18]. Moreover, an alternative solution for the occlusion problem is to model the ambiguity by predicting multiple hypotheses [3] or by predicting a probability distribution over the SMAL parameters [27].

## Acknowledgement

This research is supported in part by the National Research Foundation, Singapore under its AI Singapore Program (AISG Award No: AISG2-RP-2020-016) and the Tier 2 grant MOET2EP20120-0011 from the Singapore Ministry of Education.

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
