# Coarse-to-fine Animal Pose and Shape Estimation: Supplementary Material

**Chen Li      Gim Hee Lee**
Department of Computer Science
National University of Singapore
{lic, gimhee.lee}@comp.nus.edu.sg

We conduct further ablation studies for our approach in this supplementary material, including comparison with test-time optimization and sensitivity analysis of the refinement stage. Additional qualitative results are also provided.

**Comparison with test-time optimization.**  We compare our coarse-to-fine approach with the test-time optimization approach. As has been done in our coarse-to-fine pipeline, we also use the output from our coarse estimation stage as an initialization. Instead of apply the mesh refinement GCN, we further optimize the SMAL parameters based on the keypoints and silhouettes for 10, 50, 100, 200 iterations, respectively. We show the average PCK and IOU in the Table 1, as well as the inference time for the test set. We can see that the performance of the test-time optimization gets better with more optimization iterations. However, the inference time also increases linearly with the number of optimization iterations. In comparison, our regression based refinement achieves better performance with faster inference time. Moreover, the test-time optimization requires 2D keypoints and silhouettes, which are not always available in practice.

Table 1: Comparison with test-time optimization.

| Num Iters | time (s) | Avg PCK | IOU |
|:---:|:---:|:---:|:---:|
| 10 | 570 | 78.3 | 74.2 |
| 50 | 2561 | 79.2 | 76.1 |
| 100 | 5040 | 79.9 | 77.4 |
| 200 | 10018 | 81.7 | 79.0 |
| Ours | 64.5 | 83.4 | 81.6 |

**Ablation study on the sensitivity of the second stage to the first stage results.**  The refinement stage of our approach relies on the output of the coarse estimation stage as an initial point. We test the sensitivity of our model to the first stage results by adding Gaussian noise to the SMAL and camera parameters estimated from the coarse estimation stage, respectively. We compute the standard deviation $\sigma$ of the estimated SMAL and camera parameters over the whole dataset, and set the standard deviation of the Gaussian noise to 10%, 20%, 30% and 50% of $\sigma$. We show the results under SMAL noise and camera noise in Table 2a and 2b, respectively. We can see that our model is robust to the noise adding to the SAML parameters, and relatively sensitive to the noise adding to the camera parameter. We expect the sensitivity to the camera parameter noise because we only refine the mesh vertices in the second stage.

**Additional qualitative results.**  We show more qualitative results in Figure 1. We can see that we can get reasonable results from the SMAL-based estimations in the first stage. The coarse estimations are further improved in the refinement stage, where both keypoints and silhouettes from the rendered 3D meshes align better with ground truth annotations.

35th Conference on Neural Information Processing Systems (NeurIPS 2021), Sydney, Australia.

Table 2: Adding Gaussian noise to the estimated SMAL parameters (a) and camera parameter (b).

| (a) | | | (b) | | |
|---|---|---|---|---|---|
| SMAL Noise | Avg PCK | IOU | CAM Noise | Avg PCK | IOU |
| 0.1 | 81.3 | 78.3 | 0.1 | 82.2 | 78.3 |
| 0.2 | 79.7 | 76.7 | 0.2 | 78.6 | 75.2 |
| 0.3 | 79.0 | 75.9 | 0.3 | 75.4 | 72.9 |
| 0.5 | 78.1 | 75.4 | 0.5 | 69.7 | 68.3 |

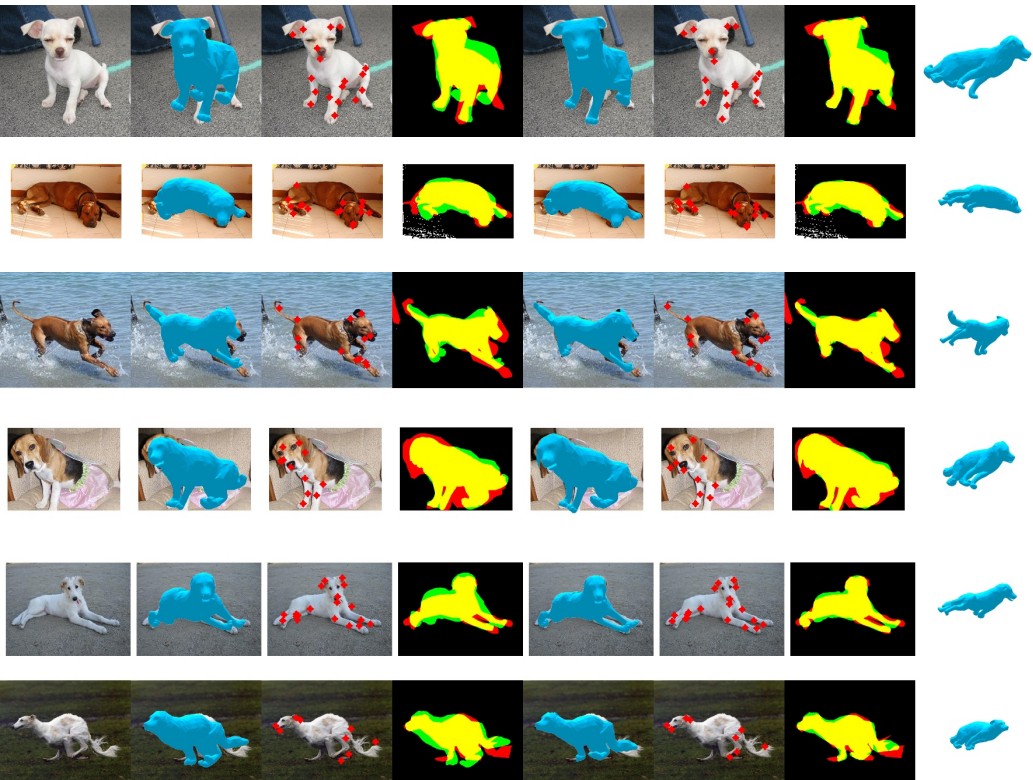

Figure 1: Some results of our coarse-to-fine approach. The first column is the input RGB images. The second to fourth columns are the estimated meshes in the camera view, the projected 2D keypoints and silhouettes from the coarse estimations. The fifth column to seventh column are the estimated meshes in the camera view, the projected 2D keypoints and silhouettes from the refined estimations. The last column is the 3D mesh in another view.