# OpenReview forum: "Coarse-to-fine Animal Pose and Shape Estimation"
_NeurIPS.cc/2021/Conference — NeurIPS 2021 Poster_

### Official Review · Reviewer_xmMe · 2021-07-14

**Rating:** 6
**Confidence:** 4

**Summary:**

- Paper focuses on 3D dog reconstruction from a single monocular image
- A coarse 3D dog estimate is generated using an existing pipeline (WLDO, SMALST), although with Tversky loss added to prevent oversized estimates (a known problem in prior work).
- The paper introduces an encoder-decoder graph convolutional network which incorporates vertex-level local image features for applying free-form refinements to the the vertices of this coarse estimate.
- These are supervised using only weak 2D supervision and are shown effective for modelling dog shapes (which are very diverse).
- Ablation study + results on StanfordExtra, BADJA and AnimalPose show state-of-the-art performance.


**Limitations And Societal Impact:**

Limitations indicated above. Societal impact is discussed briefly in conclusions.

**Main Review:**

**Strengths**

- Paper presents an effective approach to 3D dog reconstruction. This is a hard task due to shape diversity in the category, and approaches must rely on weak 2D supervision since 3D annotations are unavailable. Additionally the 3D morphable model (SMAL) is of low detail, having been constructed from scanned artist toys rather than real subjects.
- I'm not fully convinced of the novelty of all parts, but the combination of ideas (refining coarse prediction with encoder-decoder GCN that incorporates local vertex information) is novel and particularly suitable for animal reconstruction. The contribution of these components are supported by results, although only using 2D metrics.
- Results appear of high quality, and are shown on diverse subjects with a range of challenging poses. However, results are shown only from the front view.
- Tversky loss from semantic segmentation task is proposed to avoid oversized SMAL estimates -- a problem in previous work.
- Paper is well written and easy to follow. Code was provided at time of submission.

**Weaknesses**
- The novelty appears limited. [Fully Convolutional Mesh Autoencoder using Efficient Spatially Varying Kernels, Zhou et al.](https://arxiv.org/pdf/2006.04325.pdf) seem to describe an encoder-decoder graph network with up+down sampling layers for SMPL (see Figure 2 in the link). [Pixel2Mesh, Wang et al.](https://arxiv.org/abs/1804.01654) has demonstrated using local features in a GCN and [Convolutional Mesh Regression, Kolotouros et al. CVPR 2019](https://arxiv.org/abs/1905.03244) has demonstrated using a GCNs as a refinement network. Additionally, [Lions and Tigers and Bears; Zuffi et al](https://files.is.tue.mpg.de/black/papers/zuffiCVPR2018.pdf) show that using free-form vertex deformation (albeit via energy minimization) improves the accuracy of animal shape fitting. This seems to leave the novelty of this paper the combination of these techniques, their application to single-image monocular 3D dog reconstruction and supervising these architectures with weak 2D supervision only.

- The authors compare their method on numerous 2D datasets (StanfordExtra, BADJA, AnimalPose) demonstrating quant/qual improvements over state of the art approaches. However, evaluating a 3D reconstruction using 2D PCK/IOU metrics, particularly rendered from the camera viewpoint is not ideal. For this reason, it is a shame the paper does not evaluate on 3D metrics which are available in RGBD-Dog, Kearney et al. CVPR 2020.

- The paper contains only figures showing the reconstructed mesh from the camera's viewing direction. This can be misleading and causes some doubt when evaluating qualitative results. Reversed views of the mesh are needed.

- L158-163. Claim that oversized SMAL estimates caused by image foreground occupying a large area of the image after the cropping + this imbalance causes the network to estimate a mesh with larger size or larger focal length. I don't follow the logic here. The silhouette error is minimized if the predicted mesh matches the silhouette exactly. Why would preprocessing cause difficulties? However, I agree that the Tversky loss which penalizes false positive predictions, would resolve systematic issue of oversized predictions.

- L35 - Claim "we generate synthetic data for all animals, especially endangered species?" This appears untrue; the SMALST paper generates synthetic data on the Grey's zebra (an endangered zebra species) and obtains accurate results.

- Paper title is 'ANIMAL pose and shape estimation' but results are only shown on dogs. Unless results are shown on other categories, this perhaps should be 'Dog Pose and Shape estimation'.

**Minor**
- The authors refine the vertices using graph convolutions. However, another implementation (albeit to refine parameters) shown in [Human Mesh Recovery; Kanazawa et al., CVPR 2018] is to use [recurrent steps](https://github.com/nkolot/SPIN/blob/master/models/hmr.py#L140). A discussion on the differences would be useful and interesting.
- Paper shows a failure case which they explain is due to the camera looking at the back of the animal. This is a hard case, and the authors put it down to a lack of views in the training data. However, I wonder if this could be better handled in modelling the ambiguity in 3D reconstructions. Perhaps work like [Probabilistic 3D Human Shape and Pose Estimation from Multiple
Unconstrained Images in the Wild; Sengupta et al] and [3D Multibodies; Biggs et al. NeurIPS 2020] would help?
- The paper proposes the Laplacian regularizer to constrain the free-form vertex deformation step. This differs from ARAP used in [Lions \& Tigers \& Bears, Zuffi et al, CVPR 2018]. Could the authors explain the choice here?
- Is your training/test split the same as published in the WLDO Github repository?

- Some typos e.g. L144  `"global feature is FED", L158 "we RECKON" (informal language), L266 - capitalize WE. Possibly others, have not checked exhaustively.

**Review score**

This paper demonstrates an effective approach for a challenging 3D reconstruction task, focusing on an articulated category with limited training data and not a huge amount of prior work. I'm not convinced of the novelty of any of the individual components (encoder-decoder GCN, free-form refinement, local vertex features, Tversky loss all seem to be prior art). However I believe the combination of these techniques to be non-trivial, and they are shown to be effective in modelling dog shapes (which are much more diverse than humans and have only 2D supervision).

The experimental section would benefit from having side-views and evaluation on a 3D dataset (e.g. RGBD-Dog) which is now available. However, previous animal papers have not conducted a 3D evaluation so I don't think this is enough to reject this paper.

All things considered, I believe this paper is interesting and pushes the field of 3D animal reconstruction forward. I am on the fence, but overall I consider this paper just about worthy of acceptance to NeurIPS.

**Time Spent Reviewing:**

4 hrs

---

> ### Author Response · Authors · 2021-08-10
> **Our response to Reviewer xmMe.**
>
> We thank the reviewer for all the precious suggestions. Please refer to our response below.
>
> **Q1 “A combination of existing techniques”.**
>
> As also mentioned by Reviewer xmMe, the combination of all those techniques, including the coarse-to-fine reconstruction pipeline, the encoder-decoder structured GCN, the local feature extraction and the introduction of the Tversky loss, is novel although each component in our framework is not new. To the best of our knowledge, we are the first to put  these techniques together for the weakly supervised 3D animal pose and shape estimation task.
>
> **Q2 “Evaluate on the RGBD-Dog dataset [20]”.**
>
> We follow the evaluate metrics used in previous works [2, 30]. We did not compare with [20] since it mainly focuses on estimating 3D pose of dogs from depth images with 3D supervision. In contrast, we estimate 3D pose and shapes from a single RGB image with 2D weak supervision. Nonetheless, we will do the evaluation on the dataset in our final paper upon request by the reviewer. Moreover, we are unable to do the evaluation during this rebuttal period because the author is on holiday currently and is unable to respond to our email request to obtain the dataset.
>
> **Q3 “Reversed views of the mesh”.**
>
> Please refer to [this link](https://drive.google.com/file/d/1ZVn8FQADpAB4DRPWlA1LWmpoG4nEPiaZ/view?usp=sharing ) for more results with an alternative view.
>
> **Q4 “Data imbalance problem”.**
>
> We crop each image with a bounding box during data preprocessing. This causes the foreground object to occupy a larger area of the image and thus an imbalance of more foreground than background pixels. Consequently, this leads to dilations of the foreground shapes since there is a higher tendency of the network to predict any pixel as foreground from an image with large foreground. This is similar to the data imbalance in the lesion segmentation [24], where the lesion area is much smaller than non-lesion area. The data imbalance of lesion and non-lesion areas in [24] results in a network with high precision but low recall. In our case, the imbalance of the foreground object (i.e. dogs) and background pixels leads to the false positive prediction, which we mitigate with the adoption of the Tversky loss [24]. We will make this clear in the final version.
>
> **Q5 “we cannot generate data for all animals, especially endangered species appears untrue”.**
>
> The synthetic data in the SMALST paper [30] is generated from 57 images of Grevy’s zebras by applying the SMALR [31]. The synthetic data generation is a non-trivial process that requires expert knowledge on optimizing a 3D model to align with multiple images. It would take further effort to generate synthetic data for each animal category. Moreover, the SMALR might not work well for animal categories that are very different from animal categories modeled by the SMAL model. We will modify our description to make it more precise.
>
> **Q6 “Results are only shown for dogs”.**
>
> Existing datasets are abundant on dogs and scarce on other animals. Our approach is also applicable to other animal categories as long as enough images with 2D keypoints and silhouette are available for training. We can change our title to make it specific to dogs upon the request of the reviewer, albeit there is nothing in our proposed framework that limits it to only dogs.
>
> **Q7 “A discussion on the difference with recurrent refinement step in [10]”.**
>
> HMR[10] is a model-based approach. The iterative refinement steps in [10] aim to better regress the SMPL parameters because the direct regression is challenging (especially for rotation parameters). Our coarse-to-fine pipeline combines model-based and model-free representations, where the model-free refinement step aims to improve the initial model-based 3D shape. We show that this combination is able to compensate for the representation capacity of the SMAL model.
>
> **Q8 “Handle the failure case of back view images by modeling the ambiguity”.**
>
> We agree with the reviewer that it is plausible to model the ambiguity (by predicting multiple hypotheses [multibodies, Biggs et al.] or predict shape and pose distributions [Probabilistic model, Sengupta et al. ]) to improve 3D reconstructions, especially for images with occlusions. We thank the reviewer for pointing this out, and think that modeling animal pose ambiguity is a promising new research direction which we will leave for our future work.
>
> **Q9 “Laplacian prior vs arap prior”.**
>
> We have also tried the arap prior, and it does not outperform the Laplacian prior. Furthermore, arap prior requires additional steps for calculating the rigid transformation, which is more expensive than Laplacian prior. Consequently, we use the laplacian prior in our work.
>
> **Q10 “Is train/test split the same with WLDO”.**
>
> Yes, we use the same train/test split as WLDO.
>
> **Q11 “Some typos”.**
>
> Thanks for pointing this out,  we will modify it in the final version.

---

> > ### Comment · Reviewer_xmMe · 2021-08-31
> > **Rebuttal Response**
> >
> > Thank you to the authors for their detailed response, particularly for supplying images with alternative views.

---

### Official Review · Reviewer_nqoz · 2021-07-15

**Rating:** 6
**Confidence:** 5

**Summary:**

This paper studies the problem of joint animal pose and shape reconstruction from a single image. This work is largely based on the literature [Ref 2] but adopts a coarse-to-fine mechanism for joint pose and shape learning.  The method extracts global embedding as before in the first stage. In the second stage, this work leverages a local feature extractor to provide pixel-aligned local features for mesh refinement. Specifically, the extracted local features are given as input to the U-Net to refine meshes at a different resolution. Experimental comparisons with existing work [Ref 2, 3, 32] have been conducted on the StanfordExtra dataset and BADJA dataset.

**Ethical Concerns:**

N / A

**Limitations And Societal Impact:**

Yes

**Main Review:**

Strengths:
* [S1] The proposed coarse-to-fine design makes sense and the paper is generally clearly written with a good amount of technical details and experimental setups.
* [S2] The proposed method achieves superior IOU and PCK@0.15 performances compared to the previous work. The qualitative results in Figure 4 look promising.

Weaknesses:
* [W1] While the 2D projections from the input camera view look decent, it is unclear if the actual 3D shape looks reasonable or not. The reviewer would expect to see the rotated shape from another angle as did in [Ref 2] (see Figure 5 and Figure 6 of [Ref 2]). It becomes difficult to evaluate the significance of the results without such information.
* [W2] Reviewer understands that the proposed method is regression-based, but would like two see if the test-time optimization could further push the limit. In addition, when running test-time optimization with keypoints available, is this method still superior to [Ref 2]? If so, how much performance gain comes from the coarse-to-fine refinement design? It would be good to have such discussions in the rebuttal.
* [W3] The reviewer would like to know if the second stage method is very sensitive to the first stage result. In other words, how does the first stage reconstruction affect the final performance? It would be great to provide some ablation studies on poor fitting on the camera parameter and/or SMAL model parameter.


**Time Spent Reviewing:**

6

---

> ### Author Response · Authors · 2021-08-10
> **Our response to Reviewer nqoz.**
>
> We thank the reviewer for all the precious suggestions. Please refer to our response below.
>
> **Q1 “Alternative view visualization”.**
>
> Please refer to [this link](https://drive.google.com/file/d/1ZVn8FQADpAB4DRPWlA1LWmpoG4nEPiaZ/view?usp=sharing) for more results with an alternative view.
>
> **Q2 “Compare with test-time optimization”.**
>
> We compare our coarse-to-fine approach with the test time-optimization approach. For fair comparison, we use the output from our coarse estimation stage as initialization (our coarse estimation is an reimplementation of [ref2], which achieves similar performance as shown in Table 1 of our main paper), and optimize the SMAL parameters for each test image for 10, 50, 100, 200 iterations, respectively. We show the average PCK and IOU in the following Table, as well as the inference time for the test set. We can see that the performance of the test-time optimization gets better with more optimization iterations. However, the inference time also increases linearly with the number of optimization iterations. In comparison, our regression based refinement achieves better performance with faster inference time. Moreover, as the reviewer mentioned, the test-time optimization requires 2D keypoints and silhouette, which are not always available in practice.
>
> |  Num Iters   | time(s)  | PCK | IOU |
> |  :-----:  | :-----:  | :-----: | :-----: |
> | 10  | 569.60 | 78.33 | 74.24 |
> | 50  | 2561.06 | 79.21 | 76.11 |
> |  100  | 5040.34  | 79.88 | 77.38 |
> |  200  | 10018.00  | 81.68 | 79.00 |
> |  Ours  | 64.46  | 83.63 | 81.15 |
>
> **Q3 “Is the second stage sensitive to the results of the first stage”.**
>
> We test the sensitivity of our model to the first stage results by adding Gaussian noise to the SMAL and camera parameters estimated from the first stage, respectively. We compute the standard deviation $\sigma$ of the SMAL and camera parameters estimated from the dataset, and set the standard deviation of the gaussian noise to 10%, 20%, 30% and 50% of $\sigma$. We show the results in the following Tables, which show that our model is robust to the noise adding to the SAML parameters, and relatively sensitive to the noise adding to the camera parameters. We expect the sensitivity to the camera parameter noise because we only refine the mesh vertices in the second stage.
>
> |  SMAL noise   | PCK | IOU |
> |  :-----:  | :-----:  | :-----: |
> | 0.1 | 81.31 | 78.32 |
> | 0.2 | 79.72 | 76.74 |
> | 0.3 | 79.01 | 75.93 |
> | 0.5 | 78.08 | 75.42 |
>
> | CAM noise   | PCK | IOU |
> |  :-----:  | :-----:  | :-----: |
> | 0.1 | 82.16 | 78.32 |
> | 0.2 | 78.63 | 75.20 |
> | 0.3 | 75.41 | 72.85 |
> | 0.5 | 69.66 | 68.31 |

---

### Official Review · Reviewer_HKW6 · 2021-07-16

**Rating:** 6
**Confidence:** 5

**Summary:**

This paper presents a coarse-to-fine pipeline to recover animal pose and shape. They first predict SMAL parameters and then use an MRGCN to refine surface vertices to achieve better image alignment. Their MRGCN follows an encoder-decoder structure, which is the main difference against previous GCN based methods. Their method achieves state of the art results on StanfordExtra, Animal Pose and BADJA datasets.

**Limitations And Societal Impact:**

However, I feel like the qualitative results are not satisfactory, see my comments below.

## Other Questions
1.	Does this paper only estimate pose and shape of dogs? Although this paper claims ‘animal’ pose and shape estimation, I can see only dogs in Fig.1 and Fig.4. The training dataset StanfordExtra contains only dogs. I do not know their performance on other animals.
2.	In Fig.4, these dogs have very different body shapes and surface geometry, however, I find the shape results in Column 5 are quite similar. As a comparison, qualitative results in WLDO (Fig.1 of their paper) demonstrate richer shape variability. I think this is a weakness of qualitative results.
3.	I would like to see an alternative view visualization of the models in Fig.4.


**Main Review:**

## Originality
As far as I know, coarse-to-fine pipeline is very common in human pose and shape estimation area, and GCN is well investigated for human body or hand prediction. A refinement stage could certainly reduce the reconstruction error. So, the overall novelty of this paper seems low.
However, it is the first time for me to see an encoder-decoder GCN structure, and the authors prove its ability in their ablation study.

## Quality
This paper is easy to read except some spelling mistake (e.g. L.144, feed -> fed). They do a lot of experiments to show the effectiveness of each part of their pipeline. Their quantitative results achieve the best.

**Time Spent Reviewing:**

4 hours

---

> ### Author Response · Authors · 2021-08-10
> **Our response to Reviewer HKW6.**
>
> We thank the reviewer for all the precious suggestions. Please refer to our response below.
>
> **Q1  “The overall novelty of this paper seems low. However it is the first time for me to see an encoder-decoder GCN structure”.**
>
> To the best of our knowledge, we are the first to propose the weakly supervised coarse-to-fine animal pose and shape reconstruction pipeline that combines the model-based coarse estimation and model-free fine estimation. Moreover, we design an encoder-decoder structured GCN for the mesh refinement with the combination of the global and local features.  The Tversky loss is also introduced to mitigate the oversized mesh estimation problem.
>
> **Q2 “The paper is easy to read except for some spelling mistakes”.**
>
> Thanks for pointing this out, we will modify it in the final version.
>
> **Q3 “Does this paper only estimate the pose and shape of dogs”.**
>
> We only show results for the dog category because the StanfordExtra dataset is, as far as we know,  currently the only available large scale animal dataset that contains 2D keypoints and silhouette annotations. Although there are other datasets, such as Animal-pose and BADJA, they contain a limited number of images for each category. The TigDog dataset [A] contains thousands of images for horse and tiger. However, the 2D silhouettes that are estimated using [B] are very noisy. It should be noted that our approach can be applied to other animal categories when the data becomes available.
>
> [A] Luca Del Pero at el.  Articulated Motion Discovery using Pairs of Trajectories. CVPR 2015.
>
>
> [B] Luca Del Pero at el. Recovering Spatiotemporal Correspondences between Deformable Objects by Exploiting Consistent Foreground Motion in Video. arXiv 2015.
>
> **Q4 “Show more qualitative results with richer shape variability and alternative view visualization”.**
>
> Please refer to [this link](https://drive.google.com/file/d/1ZVn8FQADpAB4DRPWlA1LWmpoG4nEPiaZ/view?usp=sharing) for more results with an alternative view.

---

> > ### Comment · Reviewer_HKW6 · 2021-09-02
> > **I keep my rating**
> >
> > The rebuttal covers all my concerns. I agree with other reviewers that the novelty seems limited and experiments are constrained to dogs. If the authors could improve the experiments, I think it will be a good paper. For now, I keep my rating.

---

### Official Review · Reviewer_gzXJ · 2021-07-16

**Rating:** 7
**Confidence:** 5

**Summary:**

This paper presents a method for estimating the pose and shape of animals from images. The paper addresses limitations of previous approaches that cause misalignments and 2D keypoint locations. The main idea is to initialize using a parametric animal model and subsequently use a graph convolution network (GCN) to optimize for vertex positions to better fit shape segmentation. Experimental results how improved performance.

**Ethical Concerns:**

No.

**Limitations And Societal Impact:**

While limitations and broader impacts are discussed, these are limited to 2-3 lines. I would suggest expanding discussion of limitations, particularly discussing failure in figure 4, last row. Adding bolded subheadings would also make these sections more prominent.

**Main Review:**

I would like to begin by listing several things I like with the paper.

- The problem addressed is important and gaining interest in the community.
- The paper presents several interesting ideas and identifies the key problem with using parametric models for image-based estimation tasks. The problems stated are not only a problem for animals but also for human parametric body models and estimation for human shape/pose from images.
- The paper is well written and experiments appear to be comprehensive and convincing.

In the following, I will focus on comments, questions, issues and suggestions to improve the paper (in no particular order).

- In figure 1, it is not immediately clear that the red and green masks denote false negatives and false positives. It would help to make this clear in the caption/legend.

- Related work: There's quite a bit of previous work that has recognized the issues that the paper has identified, i.e., using 3D models for image-based estimation results in misalignments at object boundaries because the 3D models do not model the image formation process.  Here are a couple of examples.

For humans, PiFU suggests tying 3D models to image pixels. This mitigates the misalignment problem.

PIFu: Pixel-Aligned Implicit Function for High-Resolution Clothed Human Digitization, Saito et al.

More recent work proposes to model cameras explicitly (new work that appeared only recently).

CAMPARI: Camera-Aware Decomposed Generative Neural Radiance Fields, Niemeyer et al.

The paper also makes it sound like this problem is unique to animal reconstruction, but in it's really a large problem and points to important limitations of existing parametric models.

Furthermore, there is rich literature on convolutions on meshes which would be worth discussing. For instance, see:

MeshCNN: A Network with an Edge, Hanocka et al.

- The proposed "coarse" fitting method makes sense. For the mesh refinement, I would liked to see a bit more intuition for *why* the proposed GCN method is well suited? Why not some other method like predicting per-vertex correctives (as is common for faces) using a simpler neural net?

- My biggest issue with the paper is that it claims to refine both shape and keypoints, but very little details are provided for how the keypoints are refined. In figure 4, it is clear that the keypoints are improved, but *how*? There is a vague reference in lines 220, but other than that I see no other information.

- Line 221 says "estimated camera parameter f". Where is this estimated?

Misc.

- Line 137: "verctors" --> "vectors"


Overall, I believe the paper moves the state of the art forward in the field of animal shape/pose estimation, and maybe even has some lessons for human shape/pose estimation. There are some exposition and reasoning issues which I would encourage the authors to address in the rebuttal.

**Time Spent Reviewing:**

2

---

> ### Author Response · Authors · 2021-08-10
> **Our response to Reviewer gzXJ.**
>
> We thank the reviewer for all the precious suggestions. Please refer to our response below.
>
> **Q1 “Figure 1 is not clear”.**
>
> Thanks for pointing this out. we will make it clear in the final version.
>
> **Q2 “Related works such as PiFU, CAMPARI”.**
>
> PIFu introduces an implicit representation which aligns local features at the pixel level to the global context of the 3D objects. Our local feature extractor in Figure 2 plays a similar role in extracting pixel level features for refinement.  This local feature extractor is one component of our coarse-to-fine pipeline. PiFU is trained with 3D ground truth data, which is not available in our case. We thus exploit the SMAL model in the first stage to facilitate weakly supervised training with only 2D annotations. The key idea of CAMPARI is to learn a camera parameters generator jointly with the image generator for 3D-aware image synthesis so that the camera viewpoint can be explicitly controlled during synthesis.  In our work, we also estimate the camera parameters from the input image. However, our objective is to render the generated 3D mesh into the 2D image space with the estimated camera parameters to weakly supervise our network with the 2D annotations. We will include this discussion in the final version.
>
> **Q3 “The paper also makes it sound like this (misalignment) problem is unique to animal reconstruction, but it's really a large problem and points to important limitations of existing parametric models”.**
>
> We agree with the reviewer that the (misalignment) problem is not unique to the animal reconstruction task, and it is not our intention to make it sound unique. However, the problem is more severe for animal reconstruction because the SMAL model is only learned from 41 scans of toy animals. This results in the lower representation capacity of the SMAL model compared to the counterpart SMPL model, which is learned from thousands of real human scans. Our descriptions are therefore specifically targeted at the (misalignment) problem in animal pose estimation, which is also the key motivation of our paper.
>
> **Q4 “Why use the proposed GCN instead of a simpler neural net”.**
>
> A GCN is a natural choice to encode a mesh model since the mesh model is essentially a graph, where the mesh nodes and connections are the vertices and edges of the graph. Dependencies between the mesh nodes can also be effectively captured via message passings in the GCN. In contrast, it is less effective to use a simple neural net to represent the graph structure of the mesh models, and less appropriate to encode the dependencies between the mesh nodes.
>
> Furthermore, GCNs have also been successfully applied to the human pose and shape estimation task ([6] and [16] in our main paper). Nonetheless, our proposed GCN is different from the typical GCN [14] in two ways: 1) Our encoder-decoder structure enables a hierarchical multi-scale representation, and 2) the combination of global and local features allows more detailed shape reconstruction. The effectiveness of our GCN designs is evident from the ablation studies that remove the corresponding components (-ED, -LF).
>
> **Q5 “How the keypoints are improved”.**
>
> The keypoints are associated with the mesh vertices that are refined with the GCN.
>
> **Q6 “Where is the camera matrix estimated”.**
>
> The camera parameters are regressed in the first stage, which is illustrated as the ‘camera parameter’ in Figure 2 and described in line 144-145 of our main paper.
>
> **Q7 “Some typos”.**
>
> Thanks for pointing this out, we will modify it in the final version.
>
> **Q8 "Expanding discussion of limitations".**
>
> Thanks for the suggestion. We will expand the discussion of limitations in the final version.

---

### Decision · Program_Chairs · 2021-09-27

**Decision:**

Accept (Poster)

**Comment:**

The paper received 4 positive final ratings: 7, 6, 6, 6.
On the positive side, the reviewers appreciated importance of the problem, an overall meaningful approach and strong empirical performance. The main remaining concern was around novelty of individual components, but at the end the reviewers agreed that the combination of those is sufficiently novel and effective in the given setting. The remaining concerns were mostly around clarity, gaps in related works, and somewhat limited evaluation (dogs only). Some of those were addressed in the rebuttal, as acknowledged by the reviewers.
The final recommendation is to accept as a poster. The authors are highly encouraged to incorporate all feedback from the reviewers in the camera ready version of the manuscript.